

# Pseudospectral implementation of the Einstein-Maxwell system

Jorge Expósito Patiño⋆, Hannes Robert Rüter and David Hilditch

CENTRA, Departamento de Física, Instituto Superior Técnico – IST,
Universidade de Lisboa – UL, Avenida Rovisco Pais 1, 1049-001 Lisboa, Portugal

⋆ jorge.exposito.patino@tecnico.ulisboa.pt

## Abstract

Electromagnetism plays an important role in a variety of applications in gravity that we wish to investigate. To that end, in this work, we present an implementation of the Maxwell equations within the adaptive-mesh pseudospectral numerical relativity code BAMPS. We perform a thorough analysis of the evolution equations as a first order symmetric hyperbolic system of PDEs. This includes both the construction of the characteristic variables for use in our penalty boundary communication scheme, as well as radiation controlling, constraint preserving outer boundary conditions which, for the first time in a numerical context, are shown to be boundary-stable. After choosing a formulation of the Maxwell constraints that we may solve for initial data, we move on to show a suite of numerical tests. Our simulations, both within the Cowling approximation, and in full non-linear evolution, demonstrate rapid convergence of error with resolution, as well as consistency with known quasinormal decay rates on the Kerr background. Finally we evolve the electrovacuum equations of motion with strong data, a good representation of typical critical collapse runs.



# 1 Introduction

In this paper we initiate a series of numerical studies of the dynamics of the Einstein-Maxwell system. The Einstein-Maxwell system represents the interaction of electromagnetic waves and gravity. It is a realistic and consistent model that could have relevant, unexplored astrophysical applications in the strong-field regime. Moreover, we expect the model to allow us to learn about the interaction of matter and gravity in a simplified setting. Questions of interest include those concerning the effect of strong electromagnetic fields in the dynamics and relaxation of black holes, the interplay between electromagnetic and gravitational waves, including possible black hole formation from large-amplitude electromagnetic waves and, more immediately, the study of critical collapse. Particularly in the context of this last topic, where the numerical evolutions become highly challenging, a clear and robust set of code verification and validation tests is crucial. When tuned *very* close to the threshold of collapse, different methods have occasionally been seen to result in different physical results (see [1–3] for a detailed discussion). The purpose of the present paper is therefore to present a suite of verification and validation tests of our implementation so that we can confidently claim validity of our results while anticipating potential future difficulties.

Several implementations of the Maxwell equations have been employed in the context of numerical relativity. Continuum formulations of the equations have been built, for example, upon the work of [4] or [5]. Numerical simulations have been performed to understand the effect of the late-stages of inspiral and merger on the electromagnetic field with uncharged compact objects [6]. A series of works have been carried out on the effect of the electromagnetism on the inspiral, merger and stability of charged black holes [7–9]. Charged black holes have also been treated in extreme configurations [10–13] with a view to examining the effect of the Maxwell field on zoom-whirl behavior and extremality.

Here we present an implementation of the Einstein-Maxwell system inside the BAMPS code [14, 15]. The code uses the PseudoSpectral Collocation (PSC) method which has been shown empirically to result in highly accurate simulations when dealing with smooth spacetimes [16, 17]. This is expected on the basis of exponential convergence rates of the PSC, as compared with power-law convergence provided by finite difference (FD) approaches. To the best of our knowledge, this is the first PSC implementation of the Einstein-Maxwell system (although see [18] for a Cowling implementation of force-free electrodynamics). Besides the PSC method, we employ constraint damping along with radiation controlling, constraint preserving boundary conditions, both of which suppress unphysical behavior in our time evolutions. We prove boundary stability for our choice of boundary conditions, which is the first time ever that boundary stability has been shown for the Einstein-Maxwell system.

The paper is organized as follows. In section 2 we give a continuum analysis of the evolution equations, including a discussion of adjustments to the system to aid numerical evolution. We present the constraint damping scheme, the characteristic variables of the system, the formulation as an initial boundary value problem including our boundary conditions, and our method for specifying constraint satisfying initial data. In section 3 we describe the implementation in detail. A brief description of the PSC method is given, and the penalty and Bjørhus methods for the boundary conditions are outlined, as well as our symmetry reduction

(cartoon) method. In section 4 we present various tests that ensure correctness of distinct parts of the implementation. We show both verification through convergence tests and also validation by checking agreement with well established physics, such as quasinormal modes around a Kerr black hole. Furthermore, we include a simulation representative of the type of runs we will perform in the study of the threshold of black hole formation. Our conclusions are contained in section 5. We use geometric units ($G = c = 1$) and Gaussian units for the Maxwell equations ($\varepsilon_0 = 1/4\pi$).

## 2 Continuum analysis

In this section we give a continuum formulation of the Maxwell equations, compute the associated characteristic variables, demonstrate boundary stability of the system with our boundary conditions, and examine the Maxwell constraints in a form amenable to numerical solution.

### 2.1 Einstein-Maxwell system and constraint damping

The equations we want to evolve are the Einstein-Maxwell system, generally expressed as

$$R_{\mu\nu} - \frac{1}{2}Rg_{\mu\nu} = 8\pi T_{\mu\nu}, \tag{1}$$

$$\nabla_\mu F^{\mu\nu} = -4\pi j^\nu, \tag{2}$$

$$\nabla_\mu (F^*)^{\mu\nu} = 0, \tag{3}$$

$$T_{\mu\nu} = \frac{1}{4\pi}\left(F_{\mu\lambda}F_\nu{}^\lambda - \frac{1}{4}g_{\mu\nu}F_{\rho\sigma}F^{\rho\sigma}\right). \tag{4}$$

$F_{\mu\nu}$ is the Faraday tensor, $(F^*)^{\mu\nu} = -\frac{1}{2}\epsilon^{\mu\nu\rho\sigma}F_{\rho\sigma}$ is its Hodge dual and $j^\mu$ is the charge four-current density. We use the convention that $\epsilon^{0123} = -1/\sqrt{-g}$ and $\epsilon_{0123} = +\sqrt{-g}$ following [5]. The gravity-electromagnetism interaction comes both from the energy-momentum tensor for Einstein equations, which is quadratic on the Faraday tensor, as well as from the covariant derivatives on the Maxwell equations. For numerical relativity we want an initial value formulation, which leads us to define the (foliation-dependent) electric and magnetic fields

$$E^\mu = -n_\lambda F^{\lambda\mu}, \qquad B^\mu = -n_\lambda (F^*)^{\lambda\mu}, \tag{5}$$

with $n^\mu$ the vector normal to constant time slices. In terms of these, the evolution equations are [5]

$$\partial_t E^i = \mathcal{L}_\beta E^i + {}^{(3)}\epsilon^{ijk}\partial_j(\alpha B_k) + \alpha K E^i - 4\pi\alpha{}^{(3)}j^i, \tag{6}$$

$$\partial_t B^i = \mathcal{L}_\beta B^i - {}^{(3)}\epsilon^{ijk}\partial_j(\alpha E_k) + \alpha K B^i, \tag{7}$$

where we introduced the spatial metric,

$$\gamma_{\mu\nu} = g_{\mu\nu} + n_\mu n_\nu, \tag{8}$$

the lapse, $\alpha = -n_0$, the shift, $\beta^i = -\alpha n^i$, the trace of the extrinsic curvature,

$$K = -\frac{1}{2}\gamma^{ij}\mathcal{L}_n\gamma_{ij}, \tag{9}$$

the spatial charge-current density,

$$^{(3)}j^i = \gamma^i{}_\mu j^\mu, \tag{10}$$

and the timelike projection of the Levi-Civita tensor

$$^{(3)}\epsilon_{\mu\nu\lambda} = n^\sigma \epsilon_{\sigma\mu\nu\lambda}\,. \tag{11}$$

As a reminder, the Lie derivative of a vector is

$$\mathcal{L}_\beta E^i = \beta^k \partial_k E^i - E^k \partial_k \beta^i\,, \tag{12}$$

and for a scalar, it reduces to the usual directional derivative,

$$\mathcal{L}_\beta \mathcal{G}_E = \beta^k \partial_k \mathcal{G}_E\,. \tag{13}$$

Observe that, in vacuum, equations (6) and (7) have the following symmetry

$$E^i \to B^i\,, \qquad B^i \to -E^i\,, \tag{14}$$

which simplifies finding the magnetic equation once the electric one is given. In any case, here we will always present both for completeness. The Maxwell equations also provide two constraint equations that have to be satisfied in all slices of the foliation. Those are

$$\mathcal{G}_E := D_i E^i - 4\pi\rho_E = 0\,, \qquad \mathcal{G}_B := D_i B^i = 0\,. \tag{15}$$

The term $\rho$ is defined as $\rho_E = -n_\mu j^\mu$ and $D_i$ is the Levi-Civita covariant derivative of the spatial metric, defined as

$$D_i E^k = \gamma_i{}^\mu \gamma_l^k \nabla_\mu E^l\,, \tag{16}$$

for any spatial vector $E^k$. If the evolution equations are exactly satisfied, these constraints evolve as

$$\partial_t \mathcal{G}_E - \mathcal{L}_\beta \mathcal{G}_E = \alpha K \mathcal{G}_E\,, \tag{17}$$
$$\partial_t \mathcal{G}_B - \mathcal{L}_\beta \mathcal{G}_B = \alpha K \mathcal{G}_B\,. \tag{18}$$

Therefore, if the constraints are satisfied in the initial data, the constraint violation remains zero in the continuum limit. Of course, in numerical calculations a combination of truncation error and round-off introduces constraint violations. We therefore modify the equations in such a way that:

1. The equations are the same when the constraints are zero.

2. Constraint violations are suppressed in the evolution.

Following this idea various different formulations of the Maxwell equations have appeared in the literature as models for GR, see for instance [19,20]. The specific modification we employ is analogous to the usual method for the generalized harmonic gauge (GHG) or Z4 formulations of GR [21–24], divergence cleaning for MHD [25] and very similar to the approach of [3]. We add two new terms to the Maxwell equations, so that Eq. (2) becomes

$$\nabla_\mu F^{\mu\nu} + \nabla^\nu Z_E - 2\kappa n^\nu Z_E = -4\pi j^\nu\,. \tag{19}$$

This modification results in a new evolved variable $Z_E$ but has the advantage of better constraint suppressing properties. $\kappa$ is a free parameter that controls the amount of constraint damping, and in section 4.1 we compare evolutions with different values of $\kappa$. We proceed analogously with Eq. (3), adding two new terms with the equivalent variable $Z_B$ for the magnetic field. With these modifications the equations for the constraints become slightly more complicated, but in Minkowski space their evolution equations reduce to the simple form

$$\Box \mathcal{G}_E + 2\kappa \partial_t \mathcal{G}_E = 0\,, \qquad \Box \mathcal{G}_B + 2\kappa \partial_t \mathcal{G}_B = 0\,, \tag{20}$$

which is the standard equation for a damped wave. We study the constraint subsystem in generality in section 2.2.

To summarize, our final formulation of the Maxwell evolution equations is

$$\partial_t E^i = \mathcal{L}_\beta E^i - \alpha D^i Z_E + {}^{(3)}\epsilon^{ijk}\partial_j(\alpha B_k) + \alpha K E^i - 4\pi\alpha {}^{(3)}j^i\,, \tag{21}$$

$$\partial_t B^i = \mathcal{L}_\beta B^i - \alpha D^i Z_B - {}^{(3)}\epsilon^{ijk}\partial_j(\alpha E_k) + \alpha K B^i\,, \tag{22}$$

$$\partial_t Z_E = \mathcal{L}_\beta Z_E - \alpha\left(D_i E^i - 4\pi\rho\right) - 2\kappa\alpha Z_E\,, \tag{23}$$

$$\partial_t Z_B = \mathcal{L}_\beta Z_B - \alpha D_i B^i - 2\kappa\alpha Z_B\,. \tag{24}$$

## 2.2 Characteristic analysis of the Maxwell system

In this and the next subsection, we establish basic PDE properties of our formulation of the Maxwell equations. We give only a very brief overview of the underlying theory threaded with these calculations, but a detailed exposition can be found in the textbooks [26, 27] or in the excellent review article [28].

We can only hope for solutions of a numerical approximation scheme to converge to solutions of the underlying PDEs if the continuum problem is well-posed. Well-posedness is the requirement that, at least locally, the PDE problem has unique solutions that depend continuously, in a suitable sense, on the given data. The system

$$\partial_t u = A^p(x,u)\partial_p u + S(x,u)\,, \tag{25}$$

is said to be symmetric hyperbolic if there exists a symmetric, positive definite matrix $H(x,u)$, called a symmetrizer, such that $HA^p$ is symmetric for each $p$. Symmetric hyperbolic systems are naturally associated with an energy density $\mathcal{E} = u^T H u$, whose existence is a necessary and sufficient condition for the system to be symmetric hyperbolic. Subject to smoothness requirements on initial data and given coefficients, systems of PDEs that are symmetric hyperbolic have a well-posed initial value problem. We are interested in the combined Einstein-Maxwell system. Since, however, the two sets of equations are coupled only through non-principal (lower derivative) terms, and in the hyperbolic context basic PDE properties are determined by the principal part we may restrict our attention to the Maxwell equations alone, assuming that the background spacetime is sufficiently smooth. An equivalent analysis of the GHG equations of motion as implemented in BAMPS can be found in [14] (see [29] for analysis of the GHG boundary conditions). Due to this, we need to consider presently only linear systems.

We now show that our formulation of the Maxwell system is symmetric hyperbolic. We do this by finding the characteristic variables, and constructing an energy norm. Finally, we also present a characteristic analysis of the constraint sub-system.

Symmetric hyperbolic PDEs are automatically strongly hyperbolic, which implies that they have a complete set of characteristic variables for each unit spatial $s_i$. To compute the characteristic variables, we seek the left-eigenvectors of the principal symbol $P^s \equiv A^p s_p$. The principal symbol can thus be arrived at by considering the evolution equations (25), and discarding derivatives transverse to $s_p$ (from this point on denoted as *transverse*) and all non-principal terms. Given such an eigenvector $l_\lambda$ with associated eigenvalue $\lambda$, the associated characteristic variable is given by the simple dot product $l_\lambda \cdot u$. Characteristic variables have a physical interpretation as the combination of evolved variables whose solutions are, in a certain approximation, simple traveling waves of speed $\lambda$.

To compute the characteristic variables when working with tensorial fields whose principal part involves primarily the spacetime metric, it is convenient to make a $2+1$ decomposition against the vector $s^i$ by defining the projection operator

$$q_{ij} = \gamma_{ij} - s_i s_j\,. \tag{26}$$

Under this decomposition, we naturally adapt the representation of the variables to the $s_p$-direction, so that the spatial tensor components associated with directions transverse to $s_p$ coincide with the components perpendicular to $s^i$, which we denote by capital Latin indices. An index $s$ denotes contraction with $s^i$, for example $E^s = E^i s_i$. With that decomposition, the principal symbol of the equations can be represented by

$$\partial_t E^s \simeq \beta^s \partial_s E^s - \alpha \partial_s Z_E \,, \tag{27}$$

$$\partial_t B^s \simeq \beta^s \partial_s B^s - \alpha \partial_s Z_B \,, \tag{28}$$

$$\partial_t Z_E \simeq \beta^s \partial_s Z_E - \alpha \partial_s E^s \,, \tag{29}$$

$$\partial_t Z_B \simeq \beta^s \partial_s Z_B - \alpha \partial_s B^s \,, \tag{30}$$

$$\partial_t E^A \simeq \beta^s \partial_s E^A - \alpha \, {}^{(2)}\epsilon^A{}_B \partial_s B^B \,, \tag{31}$$

$$\partial_t B^A \simeq \beta^s \partial_s B^A + \alpha \, {}^{(2)}\epsilon^A{}_B \partial_s E^B \,. \tag{32}$$

The symbol $\simeq$ here denotes "equality up to non-principal terms and transverse derivatives". The 2d Levi-Civita tensor is defined as

$$ {}^{(2)}\epsilon_{\mu\nu} := s^{\lambda \, (3)}\epsilon_{\lambda\mu\nu} = n^\sigma s^\lambda \epsilon_{\sigma\lambda\mu\nu} \,. \tag{33}$$

The vectors with capital Latin indices are the transverse components to the vector $s^i$, i.e.

$$E^A = q^A{}_i E^i \,. \tag{34}$$

We have four scalar characteristic variables (the longitudinal ones) and two two-vectors (the transversal ones) that are given by

$$u_\pm^{\mathrm{long},E} = \frac{1}{\sqrt{2}} \left( \pm E^s + Z_E \right) \,, \tag{35}$$

$$u_\pm^{\mathrm{long},B} = \frac{1}{\sqrt{2}} \left( \pm B^s + Z_B \right) \,, \tag{36}$$

$$u_\pm^A = \frac{1}{\sqrt{2}} \left( \pm E^A + {}^{(2)}\epsilon^A{}_C B^C \right) \,, \tag{37}$$

with speeds $\lambda_\pm = -\beta^s \pm \alpha$, all agreeing with the speed of light in the $\pm s_p$-directions. Of these, the transversal variables agree with those presented in [5], but because we work with a different formulation, the longitudinal variables and speeds differ. The inverse transformation is given by

$$E^i = \frac{1}{\sqrt{2}} \left[ s^i \left( u_+^{\mathrm{long},E} - u_-^{\mathrm{long},E} \right) + q^i{}_A (u_+^A - u_-^A) \right] \,, \tag{38}$$

$$B^i = \frac{1}{\sqrt{2}} \left[ \left( u_+^{\mathrm{long},B} - u_-^{\mathrm{long},B} \right) - {}^{(2)}\epsilon^i{}_A \left( u_+^A + u_-^A \right) \right] \,, \tag{39}$$

$$Z_E = \frac{1}{\sqrt{2}} \left( u_+^{\mathrm{long},E} + u_-^{\mathrm{long},E} \right) \,, \tag{40}$$

$$Z_B = \frac{1}{\sqrt{2}} \left( u_+^{\mathrm{long},B} + u_-^{\mathrm{long},B} \right) \,. \tag{41}$$

The characteristic variables form a complete basis and have real characteristic speeds, which shows that the system is strongly hyperbolic, we can build an energy norm independent of $s^i$, showing that the system is also symmetric hyperbolic.

$$\begin{aligned}
\mathcal{E} :=& \left( u_-^{\mathrm{long},E} \right)^2 + \left( u_+^{\mathrm{long},E} \right)^2 + \left( u_-^{\mathrm{long},B} \right)^2 + \left( u_+^{\mathrm{long},B} \right)^2 + u_+^A u_{+A} + u_-^A u_{-A} \\
=& E^k E_k + B^k B_k + (Z_E)^2 + (Z_B)^2 \,.
\end{aligned} \tag{42}$$

Observe that this differs from the physical (ADM) energy density by constraint addition, but that it is this quantity that allows us to construct a norm in solution space as

$$\|u\| = \int_\Omega \mathcal{E}(u)\epsilon \,. \tag{43}$$

The integral is calculated over a spatial slice of constant $t$ with the appropriate induced volume form.

Besides their direct physical interpretation, the characteristic variables play an important role in the analysis of the initial boundary value problem (IBVP) of symmetric hyperbolic systems. At the boundary, each characteristic variable may be classified as either in- or outgoing, depending on the sign of the associated speed. In particular, to obtain boundary conditions that do not introduce constraint violation in the domain, we also study the characteristics of the constraint sub-system.

It turns out (see [30], or [31] for a version of the result for constrained Hamiltonian systems) that a necessary condition for strong-hyperbolicity of the full system is that the constraint subsystem itself be strongly-hyperbolic. From the results above, we know that we can find a complete set of characteristic variables for the constraint subsystem. In our case, with the constraint propagating and damping adjustments, the full constraint subsystem is

$$\partial_t Z_E - \mathcal{L}_\beta Z_E = -\alpha \mathcal{G}_E - 2\kappa \alpha Z_E \,, \tag{44}$$

$$\partial_t \mathcal{G}_E - \mathcal{L}_\beta \mathcal{G}_E = -\alpha D^2 Z_E - (D^i Z_E)(D_i \alpha) + \alpha K \mathcal{G}_E \,, \tag{45}$$

$$\partial_t Z_B - \mathcal{L}_\beta Z_B = -\alpha \mathcal{G}_B - 2\kappa \alpha Z_B \,, \tag{46}$$

$$\partial_t \mathcal{G}_B - \mathcal{L}_\beta \mathcal{G}_B = -\alpha D^2 Z_B - (D^i Z_B)(D_i \alpha) + \alpha K \mathcal{G}_B \,. \tag{47}$$

In order to find the characteristic variables, we introduce the (formal) first order reduction with the following definitions

$$\mathcal{C}_i := \partial_i Z_E \,, \qquad \bar{\mathcal{C}}_i := \partial_i Z_B \,, \tag{48}$$

which leads to two decoupled principal parts of the constraint subsystem, given by

$$\partial_t Z_E \simeq 0 \,, \tag{49}$$

$$\partial_t \mathcal{C}^A \simeq 0 \,, \tag{50}$$

$$\partial_t \mathcal{G}_E \simeq \beta^s \partial_s \mathcal{G}_E - \alpha \partial_s \mathcal{C}^s \,, \tag{51}$$

$$\partial_t \mathcal{C}^s \simeq \beta^s \partial_s \mathcal{C}^s - \alpha \partial_s \mathcal{G}_E \,, \tag{52}$$

and

$$\partial_t Z_B \simeq 0 \,, \tag{53}$$

$$\partial_t \bar{\mathcal{C}}^A \simeq 0 \,, \tag{54}$$

$$\partial_t \mathcal{G}_B \simeq \beta^s \partial_s \mathcal{G}_B - \alpha \partial_s \bar{\mathcal{C}}^s \,, \tag{55}$$

$$\partial_t \bar{\mathcal{C}}^s \simeq \beta^s \partial_s \bar{\mathcal{C}}^s - \alpha \partial_s \mathcal{G}_B \,. \tag{56}$$

The variables $Z_E$, $Z_B$, $\mathcal{C}^A$ and $\bar{\mathcal{C}}^A$ are clearly seen to be characteristic variables with zero speed. The rest are

$$c_\pm^E = \frac{1}{\sqrt{2}}\left(\pm \mathcal{G}_E + \partial_s Z_E\right) \,, \tag{57}$$

$$c_\pm^B = \frac{1}{\sqrt{2}}\left(\pm \mathcal{G}_B + \partial_s Z_B\right) \,, \tag{58}$$

with speeds $\lambda_\pm = -\beta^s \pm \alpha$. The ones with negative speed represent incoming constraint violation at the boundary. We examine constraint preserving boundary conditions in more detail in the next section.

## 2.3 Outer boundary conditions

Our numerical method uses a finite domain with a time-like boundary, meaning we do not solve the Einstein-Maxwell system as an initial value problem, but rather as an initial boundary value problem. In order to obtain a well-posed IBVP, we have to choose appropriate boundary conditions. Our guiding principle for the choice of boundary conditions is to emulate as much as possible the behavior of the initial value problem.

Given that the system is symmetric hyperbolic, an obvious choice for boundary conditions that renders the IBVP well-posed would be maximally dissipative boundary conditions [20]. The multidomain numerical approach we employ (see [32, 33] for details) uses maximally dissipative boundary conditions to carve up the computational domain into many small patches in which the incoming data of each patch corresponds to the outgoing data from its neighbors (see section 3). Unfortunately, at the outer boundary, we cannot simply employ maximally dissipative boundary conditions. The reason for this is that generic boundary conditions will not be compatible with the constraints of the theory. We thus need to consider boundary conditions in that relate properly to constraint violations.

The fact that the boundary is time-like is equivalent to the algebraic condition at the boundary

$$|\beta^s| < \alpha \,, \tag{59}$$

where, here and throughout this section, $s^i$ is the spatial normal to the boundary. The incoming variables upon which boundary conditions can be freely imposed are

$$u_-^A \,, \qquad u_-^{\text{long},E} \,, \qquad u_-^{\text{long},B} \,, \tag{60}$$

and we impose two types of boundary conditions: those that specify incoming physical data and those that ensure constraint preservation.

For the first type, we construct the Newman-Penrose scalar associated with incoming electromagnetic radiation

$$\phi_0 = F_{\mu\nu} l^\mu m^\nu = u_-^A m_A \,. \tag{61}$$

Up to non-principal terms, these characteristics do not affect the incoming constraint violation, so we are free to choose maximally dissipative boundary conditions. In particular we will fix the incoming radiation to zero, which, in terms of the physical $E^i$ and $B^i$ fields, is expressed as

$$\phi_0 \mathrel{\hat{=}} 0 \quad \Rightarrow \quad u_-^A \mathrel{\hat{=}} 0 \quad \Rightarrow \quad -E^A + {}^{(2)}\epsilon^A{}_C B^C \mathrel{\hat{=}} 0 \,, \tag{62}$$

where the symbol "$\hat{=}$" means equality at the boundary. This choice results in a difference with the initial value problem, since back-scattering from waves outside the domain is neglected. The difference, however, decreases as the outer boundary is placed farther in the asymptotically flat zone. Alternatively, we could have chosen Sommerfeld-type boundary conditions

$$(\partial_t + \partial_s)\phi_0 \mathrel{\hat{=}} 0 \,, \tag{63}$$

which also results in a well-posed problem, or perhaps even higher order derivative generalizations [34–36] thereof. This would help to reduce reflections at the boundary. We have chosen to go with condition (62) since the interpretation in terms of incoming given data is more direct.

On the other hand, we have to impose boundary conditions such that there is no incoming constraint violation through the boundary. Having studied the constraint subsystem, we know that the incoming constraint violation through a timelike boundary is given by the variables $c_-^E$ and $c_-^B$, and we want to set them to zero on the boundary. This is equivalent to the following Robin condition on the fields

$$D_i E^i - \partial_s Z_E \mathrel{\hat{=}} 0 \,, \qquad D_i B^i - \partial_s Z_B \mathrel{\hat{=}} 0 \,. \tag{64}$$

Up to non-principal terms, this is equivalent to Sommerfeld boundary conditions on $Z_E$ and $Z_B$. These are *constraint preserving*, since starting with initial data that satisfies the constraints, at the continuum level the solution of the IBVP will satisfy the constraints too. Furthermore, they are *constraint absorbing*, since (with at most this many derivatives in the conditions) they help minimize reflections of the constraints at the boundary (see [35] for further discussion).

Working with those boundary conditions and our formulation of the Maxwell equations, we show in this section that the system is boundary stable. To the best of our knowledge this is the first such proof for this specific setup. Since the system is also symmetric hyperbolic, this in turn implies that the IBVP is well-posed in a generalized sense [37–39].

In a simulation, we will generally have a certain amount of numerical error, which is characterized both by being small in comparison with the solution (at least when it first appears) and very high frequency, since the error at neighboring grid points is largely independent. If we want to make sure our implementation is valid, we have to make sure that the IBVP is robust, in the sense that small high-frequency perturbations do not cause undesired effects, especially blow-up of the solution. This is where the concept of boundary stability comes in.

In order to study the behavior of the system under small perturbations on the boundary, we allow in our analysis for small arbitrary given data in the boundary

$$\phi_0 \triangleq g_1 , \qquad c_-^E \triangleq g_2 , \qquad c_-^B \triangleq g_3 . \tag{65}$$

Roughly, a system is boundary stable if the incoming Fourier-Laplace modes are bounded by the given data at the boundary [26]. The precise definition is given in (87), once the necessary terms have been defined. The Fourier-Laplace transformation is used to transform the PDE system into an ODE system with initial data at the boundary. In the following we study how this applies to the Einstein-Maxwell case.

We want to study small, high-frequency perturbations, which allows to make a couple of simplifying approximations. Since the perturbations are small, we can study the linear system around any given solution. The Maxwell part of the equations is already linear by itself, so this is the same as fixing the background metric. On the other hand, since they are very high-frequency, we can consider the background terms constant, and neglect the forcing terms, i.e. the ones that do not contain derivatives. This is the case since for any high-frequency mode $u \sim e^{k^\mu x_\mu}$

$$\partial_\mu u \sim k_\mu u \quad \Rightarrow \quad \sqrt{\delta^{\mu\nu}(\partial_\mu u)(\partial_\nu u)} \gg u . \tag{66}$$

Under those simplifications, the system of evolution equations (21)-(24) is given by

$$\partial_t E^i = \beta^k \partial_k E^i - \alpha D^i Z_E + \alpha^{(3)}\epsilon^{ijk}\partial_j B_k , \tag{67}$$

$$\partial_t B^i = \beta^k \partial_k B^i - \alpha D^i Z_B - \alpha^{(3)}\epsilon^{ijk}\partial_j E_k , \tag{68}$$

$$\partial_t Z_E = \beta^k \partial_k Z_E - \alpha D_i E^i , \tag{69}$$

$$\partial_t Z_B = \beta^k \partial_k Z_B - \alpha D_i B^i . \tag{70}$$

For a given point in the boundary, we can choose coordinates such that the frozen metric has the form [40]

$$\mathring{g}_{\mu\nu}dx^\mu dx^\nu = -dt^2 + (dx + \beta dt)^2 + dy^2 + dz^2 , \tag{71}$$

and $x = 0$ corresponds to the boundary, with $\partial_x$ being incoming into the domain there. Furthermore, under the assumption that the boundary is timelike, we have the condition

$$|\beta| < 1 . \tag{72}$$

Generally, the form of our system is

$$\partial_t u = A^p \partial_p u , \qquad B^p \partial_p u + C u \triangleq L g , \tag{73}$$

the first equation representing the evolution, and the second one the boundary conditions. Since boundary conditions are only imposed on the incoming characteristics, which in our case represent half of the degrees of freedom, $B^p$ and $C$ are $4 \times 8$ matrices, and $g$ has 4 components. Observe that we allow a general linear operator $L$ on the given data. In boundary conditions that contain up to one derivative of the solution, we allow up to one derivative, within the boundary, of the given data within this operator. This affects the specific form of the norms that are obtained within the well-posedness result (see [40] for a detailed explanation.)

By performing a Laplace-transform in time and a Fourier-transform in the space tangential to the boundary

$$\hat{f}(x, s, \omega_y, \omega_z) := \int f(t, x, y, z) e^{-i\omega_y y - i\omega_z z - st} \, dt \, dy \, dz, \tag{74}$$

we can transform (73) into an ODE system with initial data at the boundary.

The transformation of system (73) is

$$\partial_x \hat{u} = M\hat{u}, \qquad N_1 \partial_x \hat{u} + N_2 \hat{u} \,\hat{=}\, \hat{L}\hat{g}. \tag{75}$$

For the Einstein-Maxwell system, with an appropriate ordering of the variables in the solution vector, given by

$$\hat{u} = \begin{bmatrix} \hat{E}^x \\ \hat{B}^y \\ \hat{E}^z \\ \hat{Z}_E \\ \hat{B}^x \\ \hat{E}^y \\ \hat{B}^z \\ \hat{Z}_B \end{bmatrix} = \begin{bmatrix} \hat{v}_E \\ \hat{v}_B \end{bmatrix}, \tag{76}$$

with $\hat{v}_E$ and $\hat{v}_B$ vectors with four components, we can split the whole system into two identical copies. In these variables, equations (75) are of the form

$$\partial_x \begin{bmatrix} \hat{v}_E \\ \hat{v}_B \end{bmatrix} = \begin{bmatrix} \bar{M} & 0 \\ 0 & \bar{M} \end{bmatrix} \begin{bmatrix} \hat{v}_E \\ \hat{v}_B \end{bmatrix}, \tag{77}$$

$$\begin{bmatrix} \bar{N}_1 & 0 \\ 0 & \bar{N}_1 \end{bmatrix} \partial_x \begin{bmatrix} \hat{v}_E \\ \hat{v}_B \end{bmatrix} + \begin{bmatrix} \bar{N}_2 & 0 \\ 0 & \bar{N}_2 \end{bmatrix} \begin{bmatrix} \hat{v}_E \\ \hat{v}_B \end{bmatrix} \,\hat{=}\, \begin{bmatrix} \bar{L} & 0 \\ 0 & \bar{L} \end{bmatrix} \begin{bmatrix} \hat{g}_E \\ \hat{g}_B \end{bmatrix}. \tag{78}$$

This allows us to study just one half of the system. Showing boundary stability of one of the two copies will automatically demonstrate boundary stability of the whole system. The matrices given above are

$$\bar{M} = \frac{1}{1 - \beta^2} \begin{bmatrix} -s\beta & -i\beta\omega & -i\omega & -s \\ -i\beta\omega & -s\beta & s & i\omega \\ i\omega & s & -s\beta & -i\beta\omega \\ -s & -i\omega & -i\beta\omega & -s\beta \end{bmatrix}, \tag{79}$$

and

$$\bar{N}_1 = \begin{bmatrix} 0 & 0 & 0 & 0 \\ -1 & 0 & 0 & -1 \end{bmatrix}, \tag{80}$$

$$\bar{N}_2 = \begin{bmatrix} 0 & -1 & 1 & 0 \\ 0 & 0 & -i\omega & 0 \end{bmatrix}, \tag{81}$$

$$\bar{L} = \begin{bmatrix} 1 & 0 \\ 0 & s \end{bmatrix}. \tag{82}$$

The boundary conditions can be transformed into a purely algebraic equation by substituting the equation for the $x$ derivatives

$$\bar{N}_1 \bar{M} \hat{v}_E + \bar{N}_2 \hat{v}_E \triangleq \bar{L} \hat{g}_E \quad \Rightarrow \quad \bar{L}^{-1}(\bar{N}_1 \bar{M} + \bar{N}_2) \hat{v}_E = \bar{X} \hat{v}_E \triangleq \hat{g}_E, \tag{83}$$

which for the Einstein-Maxwell case results in

$$\bar{X} = \begin{bmatrix} 0 & -1 & 1 & 0 \\ \frac{1}{1-\beta} & \frac{i\omega}{s(1-\beta)} & \frac{i\omega\beta}{s(1-\beta)} & \frac{1}{1-\beta} \end{bmatrix}. \tag{84}$$

As we can see, this matrix is not square, and in particular it is not injective, so we cannot find the full solution at the boundary solely given by the boundary data. This is not an accident of the Einstein-Maxwell case. Indeed, it will happen in general, since the boundary conditions only give data for incoming characteristics variables. Physically, only half of the degrees of freedom are incoming and are specified by the given data at this boundary.

We can restrict our analysis to the variables that do depend on the given data by looking at the eigenvectors of $\bar{M}$ with negative real part eigenvalues. In general, the solutions of

$$\partial_x \hat{v}_E = \bar{M} \hat{v}_E,$$

are of the form

$$\hat{v}_E = R^{(-)} \sigma_-(x, s, \omega) + R^{(+)} \sigma_+(x, s, \omega), \tag{85}$$

where $R^{(+)}$ is the matrix of eigenvectors whose eigenvalues have positive real part, $R^{(-)}$ consists of those whose eigenvalues have negative real part and $\sigma_\pm$ are some vectors. Generally we could have eigenvectors with zero real part, in which case we would say that the boundary is characteristic with respect to some of the variables, but in our case the assumption of time-like boundary ensures that will not be the case. By restricting to the subspace of negative real part eigenvalues, the matrix $X$ becomes injective. We can find the relation of the solution to the given data at the boundary by

$$\hat{v}_E \triangleq R^{(-)} \left( X R^{(-)} \right)^{-1} \hat{g}_E =: F \hat{g}_E. \tag{86}$$

The condition of boundary stability is that there exists a bound

$$|\hat{v}_E(0, s, \omega)| < \delta |\hat{g}_E(s, \omega)|, \tag{87}$$

for some $\delta > 0$. In this case, showing boundary stability is equivalent to showing that the matrix $F$ relating the physical and given data is bounded.

In the Einstein-Maxwell case, the matrix $M$ has two distinct eigenvalues, given by

$$\tau_\pm = -\gamma^2(s\beta \mp \lambda), \tag{88}$$

where we have defined

$$\gamma := (1 - \beta^2)^{-1/2}, \qquad \lambda := \sqrt{s^2 + \gamma^{-2}\omega^2}. \tag{89}$$

These eigenvalues appear also in the general relativity case, and it has been shown [41] that

$$\text{Re}(\tau_-) < 0 < \text{Re}(\tau_+), \quad \text{given} \quad \text{Re}(s) \geq 0, \quad |\beta| < 1, \tag{90}$$

therefore the incoming variables are the ones associated with $\tau_-$. The incoming part of the solution is

$$\hat{v}_E = \underbrace{\begin{bmatrix} \beta\omega^2 + s\lambda & -i\omega(s\beta - \lambda) \\ i\omega(s\beta - \lambda) & -(\beta\omega^2 + s\lambda) \\ 0 & s^2 + \omega^2 \\ s^2 + \omega^2 & 0 \end{bmatrix}}_{R^{(-)}} \begin{bmatrix} \sigma_1 \\ \sigma_2 \end{bmatrix} e^{\tau_- x}. \tag{91}$$

We want to show a bound of $\hat{v}_E$, and for that we show that the absolute value of all matrix elements are bounded. To do so it is useful to introduce the following variables

$$k = \sqrt{|s|^2 + \omega^2}, \tag{92}$$

$$s' = s/k, \tag{93}$$

$$\omega' = \omega/k, \tag{94}$$

$$\lambda' = \lambda/k, \tag{95}$$

$$\tau' = -\tau_-/k, \tag{96}$$

in such a way that all the primed variables have bounded absolute value. With those variables we can rewrite the matrix relating given data to physical fields at the boundary as

$$F = -\frac{1}{(s'+\lambda')^2} \begin{bmatrix} \frac{i\omega'(\lambda'+s')}{(\beta+1)\gamma^2} & \frac{s'(\lambda'+s')}{(\beta+1)\gamma^2} \\ \frac{(\lambda'+s')(\beta\lambda'+s')}{(\beta-1)(\beta+1)^2\gamma^2} & \frac{is'\omega'}{(\beta+1)^2\gamma^4} \\ -\frac{(\lambda'+s')(\lambda'+\beta s')}{(\beta-1)(\beta+1)^2\gamma^2} & \frac{is'\omega'}{(\beta+1)^2\gamma^4} \\ 0 & \frac{s'(\lambda'+s')}{(\beta+1)\gamma^2} \end{bmatrix}. \tag{97}$$

It is well known [27, 29, 40] that the combination $s' + \lambda'$ is bounded below by a positive value, but for the sake of completeness, we show it here.

If $\omega = 0$, then

$$|s' + \lambda'| = \frac{|s + |s||}{|s|} = |e^{i \arg s} + 1| = \cos\left(\frac{\arg s}{2}\right) \geq \sqrt{2}, \quad \text{for} \quad \text{Re}(s) \geq 0. \tag{98}$$

On the other hand, if $\omega \neq 0$ we introduce $\zeta = s/\omega$

$$s' + \lambda' = \frac{\zeta + \sqrt{\zeta^2 + \gamma^{-2}}}{\sqrt{|\zeta|^2 + 1}}. \tag{99}$$

Assuming there is no such bound, there must exist a sequence $\zeta_n \to \zeta_*$ in the half complex plane $\text{Re}(\zeta) \geq 0$ such that

$$\frac{\zeta_n + \sqrt{\zeta_n^2 + \gamma^{-2}}}{\sqrt{|\zeta_n|^2 + 1}} \to 0 \quad \Rightarrow \quad \zeta_* + \sqrt{\zeta_*^2 + \gamma^{-2}} = 0 \quad \Rightarrow \quad \zeta_*^2 = \zeta_*^2 + \gamma^{-2} \quad \Rightarrow \quad \gamma^{-2} = 0, \tag{100}$$

and since $\gamma^{-2} \neq 0$ when the boundary is timelike ($|\beta| \leq 1$), we reach a contradiction, thus a positive lower bound must exist.

Since the matrix entries are expressed in terms of primed variables, which are bounded, and $|s' + \lambda'|$ is bounded below by a positive value, the matrix is bounded element-wise. Therefore, the solution at the boundary is bounded by the given data, which in this half of the system is

$$|\hat{v}_E(0, s, \omega)| \leq \delta |\hat{g}_E(s, \omega)|, \tag{101}$$

for some $\delta > 0$. Since the two halves of the system are the same, this bound applies to the whole system, and we have boundary stability. As mentioned before, since the system is symmetric hyperbolic, the theorems of [37–39] ensure with sufficient smoothness that the system is well-posed in the generalized sense.



## 2.4 Initial data

In order to run free-evolution simulations we need constraint satisfying initial data, meaning that we have to solve the constraint equations (15) in a curved background at the same as solving the gravity constraints. The full system of constraints is

$$\mathcal{H} = {}^{(3)}R + K^2 - K_{ij}K^{ij} - 16\pi\rho = 0\,, \tag{102}$$

$$\mathcal{M}^i = D_j\left(K^{ij} - \gamma^{ij}K\right) - 8\pi S^i = 0\,, \tag{103}$$

$$\mathcal{G}_E = D_i E^i - 4\pi\rho_E = 0\,, \tag{104}$$

$$\mathcal{G}_B = D_i B^i = 0\,, \tag{105}$$

where

$$\rho = \frac{1}{8\pi}\left(E^2 + B^2\right)\,, \tag{106}$$

$$S_i = \frac{1}{4\pi}{}^{(3)}\epsilon_{ijk}E^j B^k\,, \tag{107}$$

are the contractions of the energy momentum tensor that appear in a $3+1$ decomposition.

The constraints constitute an underdetermined system of equations, so we have to make choices for some of the variables and then solve the constraint equations for the rest. We choose to do so with the decomposition of the electromagnetic fields given by

$$E_i = \psi^{-2}\left(\tilde{E}_i - \partial_i\varphi_E\right)\,, \tag{108}$$

$$B_i = \psi^{-2}\left(\tilde{B}_i - \partial_i\varphi_B\right)\,, \tag{109}$$

where $\psi$ is the conformal factor to be found by solving the Hamiltonian and momentum constraints, $\tilde{E}^i$ and $\tilde{B}^i$ are freely specifiable variables, and $\varphi_E$, $\varphi_B$ are the variables to solve for. That decomposition reduces the constraints to Poisson-type equations on the conformal geometry

$$\bar{D}^2\varphi_E = \bar{D}_k\tilde{E}^k - 4\pi\psi^6\rho_E\,, \tag{110}$$

$$\bar{D}^2\varphi_B = \bar{D}_k\tilde{B}^k\,. \tag{111}$$

A conformal decomposition was already used in [1–3, 5]. This constraint formulation for the electromagnetic fields has a couple of advantages when paired with a conformal decomposition of the gravity variables, particularly in the electrovacuum case. Since in electrovacuum the constraints (110) and (111) depend only on the conformal geometry, they can be solved independently, and in many cases (in particular under the assumption of conformal flatness) analytical solutions already exist. Presently we use analytical solutions of the conformal constraints. These solutions have the added advantage of an easy physical interpretation as non-linear extensions of the corresponding linear solutions. This extension from linear to non-linear setting is non-unique, but we nevertheless refer to electromagnetic data that is conformally a linear quadrupole as a non-linear quadrupole, for example. On the gravity side, we use the CTS approach [42].

## 3 Implementation

The Maxwell equations are implemented within the existing numerical relativity code BAMPS, and they share the code structure and tools with the existing code base. Below we give a more

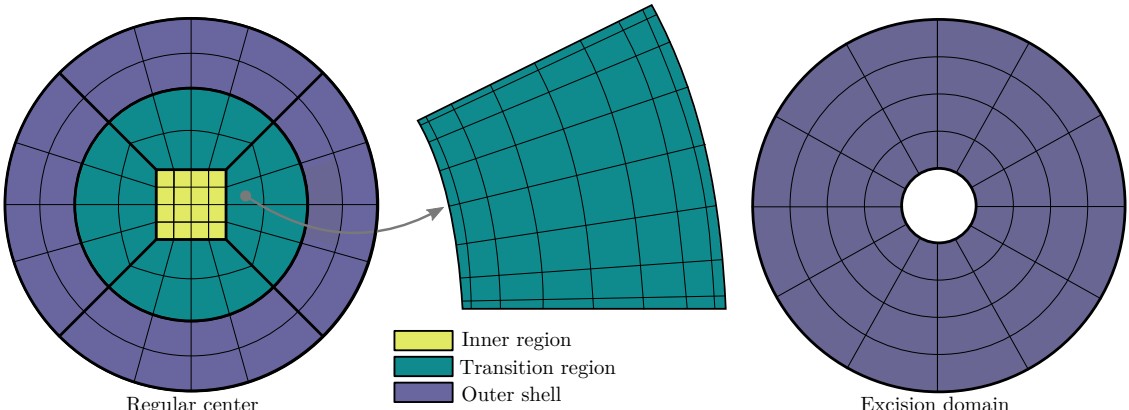

Figure 1: In BAMPS, the domain is divided into three subdomains of different geometry. Each one of them is divided in patches. Inside each patch, the solution is represented as a linear combination of Chebyshev polynomials. The number of patches in the inner region is given to match the angular resolution outside.

detailed discussion of details that relate directly to the Maxwell implementation but, for completeness, we begin with an overview of the general code structure. BAMPS is a 3d MPI parallel multidomain (spectral element) adaptive pseudospectral code taylored to the solution of first order symmetric hyperbolic PDEs. We employ the method of lines for time evolution, and the computational domain is divided into small patches, called spectral elements, on which an initial boundary value problem is solved. These elements can then be distributed to different computational cores, provided that appropriate data from neighboring elements are communicated. Since such data have to be provided on domains of codimension 1 within the computational domain, this approach is expected to scale well, which has been confirmed in earlier work, where the code has been shown to scale up to at least several thousand computational cores. That said, the experiments we have performed for this paper, in which we are primarily focused on establishing the validity of our Maxwell implementation, we used at most 768 cores. Data are communicated via a penalty method on characteristic variables. Various different options are possible at the outer boundary of the domain. When evolving black hole spacetimes we use black hole excision. The idea is that since the black hole boundary is pure outflow from the point of view of the PDEs, no continuum boundary conditions are required. We therefore simply monitor the boundary to be sure that it indeed remains outflow and apply no boundary conditions. To treat situations with symmetry we employ the cartoon method, in which a Killing vector associated with the symmetry is used reduce the dimensionality of the problem. Our mesh-refinement driver supports both $h$ and $p$ refinement. For $h$ refinment, the number of spectral elements is increased, whereas for $p$ refinement, the number of grid-points per spectral element is increased. For a full description of the code, see [14, 15].

The discretization method is a multi-patch PseudoSpectral Collocation (PSC) method. A schematic of a typical domain is presented in Fig. 1. Here we give a very abridged presentation of the PSC method (for more detail see [43, 44].) Within each patch the approximate solution is represented as a linear combination of basis polynomials. Given a general function, there are many ways to approximate it by polynomials, and the method we choose is to have the function and its approximation agree in a discrete set of points, in some way "sampling" the approximated function. Those points are called the *collocation points*. In each patch, the solution is represented as

$$f(x) \approx \sum_{i=0}^{N} f_i T_i(x), \quad T_i(x) \text{ is a polynomial of degree } i, \tag{112}$$

therefore, the derivative is approximated by

$$f'(x) \approx \frac{d}{dx} \sum_{i=0}^{N} f_i T_i(x) = \sum_{i,j}^{N} D_i{}^j f_j T_i(x),$$ (113)

where $D_i{}^j$ represents the components of a dense $N + 1 \times N + 1$ matrix. From this we can infer two important properties of the PSC method:

1. For a given number of basis polynomials, the derivative looks formally like an $N$-th order finite difference. Heuristically, every time we increase $N$ we not only make the grid spacing smaller, but increase the order of convergence, and therefore we may expect exponential convergence in spatial derivatives.

2. The derivative at a single point on the domain depends on the value on all other points in the patch. As a result, changing the boundary points by hand may have unintended effects everywhere else in the patch within one time-step. This leads us to consider an alternative way of implementing boundary conditions, which we discuss next.

Regarding the question of boundary conditions, we impose Dirichlet conditions on the incoming physical degrees of freedom (a special case of maximally dissipative conditions) and constraint preserving Robin-like conditions at the outer boundary. Besides this, since we split the domain into spectral elements, we impose Dirichlet conditions on all incoming variables at these patch boundaries to match the solutions there. The patching conditions are imposed through the penalty method [32], whereas the Robin (and Von Neumann if we had any) are imposed through the Bjørhus method [45]. Both work in a similar way: instead of modifying field values at the boundaries, the evolution equations for the incoming characteristic variables are modified. This is a weaker imposition of boundary conditions, but it is informed by studying the evolution of energy estimates. In simplified and tractable cases, it has been proven that it leads to a discrete analog of well-posedness of the initial boundary value problem.

The penalty method consists of modifying the right-hand side of the evolution of a given characteristic variable $u$ in the boundary as follows

$$\partial_t u \doteq (\partial_t u)_{\mathring{\Omega}} + p\lambda(u - u_{BC})\Theta(-\lambda).$$ (114)

$\Omega$ is the domain, $\mathring{\Omega}$ its the interior and an index $\mathring{\Omega}$ denotes the use of the same expression used in the interior. Here $\lambda$ is the speed of the characteristic, and the factor of $\Theta(-\lambda)$ (the Heaviside step function) ensures we only add modifications to the incoming characteristic variables. Finally $p$ is a free parameter fixed by energy conservation arguments on the numerical approximation, and $u_{BC}$ is the value that we want to set the characteristic to at the boundary. In patch boundaries, $u_{BC}$ corresponds to the value in the neighboring patch. As discussed in section 2.3, in the outer boundary we want to set the incoming Newman-Penrose scalar for the electromagnetic field to be zero. Equivalently, in terms of the characteristic variables we want to set

$$u_-^A \doteq 0,$$ (115)

which we achieve by the following implementation of the penalty method

$$\partial_t u_-^A \doteq (\partial_t u_-^A)_{\mathring{\Omega}} - p(\beta^s + \alpha)u_-^A.$$ (116)

We turn our attention now to constraint preserving boundary conditions. Following [46], we can set the desired Robin-type boundary conditions on the characteristic variables $u_-^{\text{long},E}$, $u_-^{\text{long},B}$ since

$$c_-^E \simeq \partial_s u_-^{\text{long},E}, \qquad c_-^B \simeq \partial_s u_-^{\text{long},B}.$$ (117)

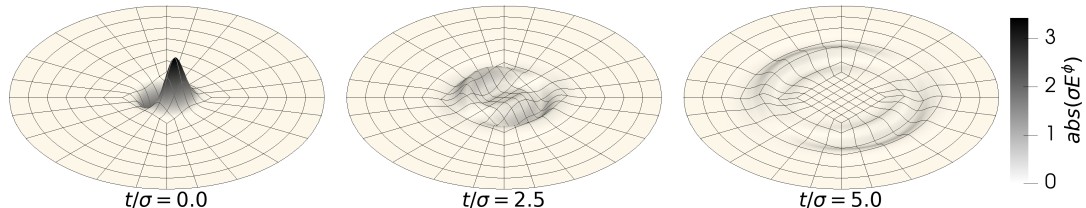

Figure 2: Snapshots of $\sigma E^{\phi}$ in the $xz$ plane at different times. We can see the expected behavior of propagation of a dipole wave. This is only a subset of the simulation domain; the true boundary is placed at double the radius of the plot.

The Bjørhus method consists of modifying the evolution equation of a characteristic variable in the following way

$$\partial_t u \mathrel{\hat=} (\partial_t u)_{\mathring{\Omega}} + \lambda \left(\partial_s u - \partial_s u_{BC}\right)\Theta(-\lambda). \tag{118}$$

Notice the differences with equation (114). Here we use the longitudinal derivatives and there is no $p$ factor.

In our case, the Bjørhus method for the constraints reduces to

$$\partial_t u_-^{\text{long},E} \mathrel{\hat=} (\partial_t u_-^{\text{long},E})_{\mathring{\Omega}} - (\beta^s + \alpha)c_-^E, \tag{119}$$

$$\partial_t u_-^{\text{long},B} \mathrel{\hat=} (\partial_t u_-^{\text{long},B})_{\mathring{\Omega}} - (\beta^s + \alpha)c_-^B. \tag{120}$$

In many cases, we want to simulate axisymmetric systems. In those cases, we can use the cartoon method [47] as implemented in [48], which allows us to reduce the simulation domain to a single $\varphi = $ constant slice. The idea is that, in axisymmetry, all tensors in the problem have the following symmetry

$$\mathcal{L}_{\partial_{\varphi}} T = 0. \tag{121}$$

Expressing that condition on a coordinate basis, we obtain formulae for the derivatives perpendicular to the domain in terms of the derivatives inside the domain. As an example, when $T$ is a vector, and considering that we take the slice $y = 0$ in Cartesian coordinates, the formula above results in the following condition for the $x$ component

$$x\partial_y v^x - y\partial_x v^x + v^y = 0 \quad \Rightarrow \quad \partial_y v^x = -\frac{1}{x}v^y. \tag{122}$$

The last point to discuss about the implementation is the initial data solver. We use the hyperbolic relaxation method of [49], which is analogous to traditional parabolic relaxation solvers, but it is more readily implemented in an evolution code like BAMPS.

## 4 Numerical results

In the experiments presented here, we aim to showcase both the correctness of the numerical implementation by demonstrating convergence, as well as the code's ability to reproduce the correct physics by comparing its results with well understood systems. For that purpose we perform the following tests:

- Test convergence to an analytical solution of electrodynamics on a flat background.

- Verify that Reissner-Nordström black hole initial data lead to a static simulation.

- Compute quasinormal mode frequencies of an electromagnetically excited black hole that relaxes to Kerr.

- Run a simulation representative of a critical collapse bisection study.

Some relevant parameters for each simulation are given in table 1. The computational domain is represented in Fig. 1. The approximate costs in core hours of the different tests are

- Eight core hours in average per simulation in the flat electrodynamics tests.

- 750 core hours for the static Reissner-Nordström test.

- 7500 core hours for the relaxation of an electromagnetically excited black hole test.

- 250 core hours for the critical-collapse-like simulation.

Table 1: Numerical parameters of the simulations in the test. There are no data on the inner and transition regions of the static Reissner-Nordström and black hole relaxation tests since we use excision, which does not have those regions (see Fig. 1).

| | Flat dipole | $\kappa$ comparison | Dynamical dipole |
|---|---|---|---|
| Angular resolution | 6 | 8 | 8 |
| Transition region resolution | 6 | 6 | 12 |
| Outer shell resolution | 6 | 4 | 10 |
| Number of collocation points | 11-21 | 15 | 15 |
| AMR | off | off | on |
| Inner region size$\times\sigma^{-2}$ | $2 \times 2$ | $2 \times 2$ | $10 \times 10$ |
| Transition region radius$\times\sigma^{-1}$ | 10 | 10 | 10 |
| Outer shell radius$\times\sigma^{-1}$ | 20 | 20 | 20 |
| Constraint transport | off | on | on |
| Damping parameter ($\sigma\kappa$) | 0 | 0-2.5 | 1 |

| | Reissner-Nordström static | Black hole relaxation |
|---|---|---|
| Angular resolution | 6 | 8 |
| Outer shell resolution | 100 | 500 |
| Number of collocation points | 11-21 | 15 |
| AMR | off | off |
| Outer shell radius$\times M^{-1}$ | 1.7 to 201.7 | 1.9 to 201.9 |
| Constraint transport | on | on |
| Damping parameter ($M\kappa$) | 1 | 1 |

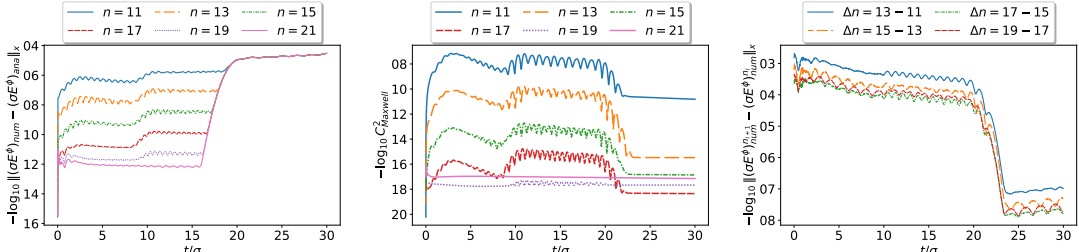

Figure 3: Convergence of various error norms as function of time $t$ with the number of collocation points $n$ for a simulation of an electromagnetic dipole wave on a flat background. Left: Norm of the difference to the analytic solution to the Cauchy problem. Observe that here it is expected, and observed, that once the outer boundary has a significant effect on the numerical solution we can no longer see convergence to a solution of the Cauchy problem. Middle: Maxwell constraint norm. We observe here that the constraints continue to converge even beyond $t/\sigma \simeq 20$, in line with our interpretation of the left panel. Right: Norm of the difference between subsequent resolutions. The norms of the differences are calculated in the $x$ axis, as in equation (133), whereas the constraint monitor is calculated with a norm over the whole domain as defined in Eq. (134).

## 4.1 Flat background electrodynamics

This test has the advantages that it only involves the newly implemented code, and we can compare the simulation results to analytical solutions. We choose dipole initial data with a Gaussian profile, which has previously been used by [2],

$$E^i \partial_i = 8 \mathcal{A} e^{-(r/\sigma)^2}(-y\partial_x + x\partial_y), \tag{123}$$

$$B^i = 0, \tag{124}$$

where $\mathcal{A}$ is the amplitude of the Gaussian profile and $\sigma$ is its width. We choose $\sigma$ to be the characteristic length scale of the system, and all values presented are in units of $\sigma$. The analytical solution that we compare to is

$$E^i \partial_i = A f_3(u, v) \frac{\partial_\phi}{\sigma}, \tag{125}$$

$$B^i \partial_i = A \frac{z}{r} [6 f_1(u, v) - f_2(u, v)] \partial_r + A [f_2(u, v) - 2 f_1(u, v)] \partial_z, \tag{126}$$

with the following auxiliary variables

$$u := \frac{r-t}{\sigma}, \qquad v := \frac{r+t}{\sigma}, \tag{127}$$

$$f_1(u, v) := \frac{u e^{-u^2} - v e^{-v^2}}{(r/\sigma)^2} + \frac{1}{2} \frac{e^{-u^2} - e^{-v^2}}{(r/\sigma)^3}, \tag{128}$$

$$f_2(u, v) := 2 \frac{e^{-u^2} - e^{-v^2} - 2u^2 e^{-u^2} + 2v^2 e^{-v^2}}{r/\sigma}, \tag{129}$$

$$f_3(u, v) := 2 \left( \frac{t_0}{\sigma} \frac{-e^{-u^2} + e^{-v^2}}{(r/\sigma)^3} + 2 \frac{u^2 e^{-u^2} + v^2 e^{-v^2}}{(r/\sigma)^2} \right). \tag{130}$$

$$\tag{131}$$

We represent the solution by the scalar

$$E^\phi := E^\mu (\partial_\phi)_\mu \, , \tag{132}$$

where $\partial_\phi$ is the angular Killing vector. Defining it this way allows us to compare with the more general case presented in section 4.4. Fig. 2 shows snapshots of $E^\phi$ in this simulation. To test convergence, we fix the number and layout of the patches, but vary the number of collocation points. For this test, we do not use constraint damping. The relevant parameters are presented in TABLE 1. Figure 3 shows the convergence to the analytical solution $E^\phi_{\text{ana}}$, the convergence of the constraint norm $C^2_{\text{Maxwell}}$ (see Eq. (134)), and the self convergence of the numerical approximate solution $E^\phi_{\text{num}}$ with respect to the number of collocation points. The norms of differences of two functions are calculated in post-processing along the $x$-axis as

$$\|f - g\|_x = \sqrt{\int_{x \text{ axis}} [f(x,0,0) - g(x,0,0)]^2 \, dx} \, , \tag{133}$$

whereas the norms of the constraints are calculated during the simulation over the entire domain.

The results are consistent with exponential convergence up to the resolution floor. At high enough resolution, we might expect the error to be dominated by the polynomial convergence of the RK4 time integrator, however our results indicate that behavior does not manifest before the resolution floor is reached. It is worth noting that, once the pulse reaches the outer boundary, located at $r = 20\sigma$, the simulation stops converging to the analytical solution. This is to be expected, since our boundary condition of no incoming radiation only approximates the solution of the initial value problem. In the self-convergence test, we find that the solution still converges after that time is reached.

Furthermore, we observe that the quantities reach a resolution floor. This can be seen in the $n = 21$ line of the convergence of the Maxwell constraint monitor defined in Eq. (134), or in the $\Delta n = 19 - 17$ line of the self-convergence plot. The exponential convergence of the PSC method, along with its higher arithmetic error associated with the dense matrix multiplications, means that reaching the floor of resolution is common. Nevertheless, one of the strengths of the approach is that the error could be reduced even further by increasing the number of spectral elements. In Fig. 4, we show a complementary test in which the number of patches is scaled up instead of the number of collocation points. The result is compatible

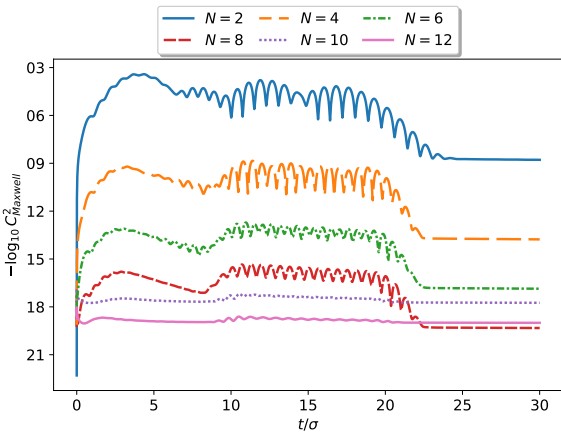

Figure 4: Comparison of the constraint monitor for different number of patches, scaled by $N^2$ in all sub-domains. The result is consistent with high-order polynomial convergence, and the most accurate run shows less error than the ones in Fig. 3.

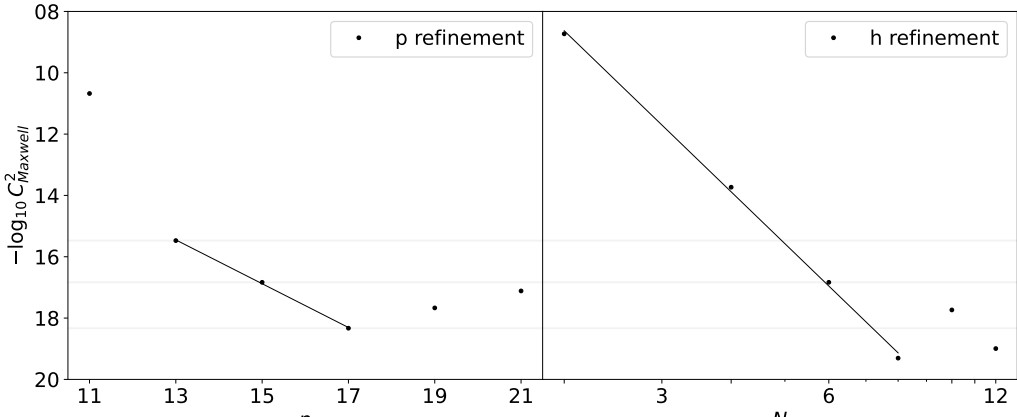

Figure 5: Constraint violation at time $t/\sigma = 25$, for both convergence tests, increasing the number of collocation points (p refinement) and increasing the number of patches (h refinement). Notice that the bottom axis scale is linear in the number of collocation points $n$ (left) and logarithmic in the number of patches $N$ (right).

with high order polynomial convergence (we are using 15 collocation points, so we expect the error to decrease with a factor of resolution to the fifteenth power, up until the time integrator error starts to dominate). Notably, in this figure we can see that the most precise run has smaller error than the one presented in Fig. 3, showing that we can decrease the error by appropriately choosing both the number of collocation points and the layout and number of patches. This can also be seen in Fig. 5, which shows the constraint violation at a late time ($t = 25\sigma$) as a function of resolution for both test cases. That figure also shows fits to the expected exponential and polynomial convergence, respectively.

We can also change the constraint damping parameter and see how the error is effected. In this case the error is monitored as the sum of the $L^2$-norms of all the constraints over the whole domain

$$
\begin{aligned}
C^2_{\text{Maxwell}} := & \lambda^{-1} \left( \|Z_E\|^2 + \|Z_B\|^2 \right) + \lambda \left( \|\mathcal{G}_E\|^2 + \|\mathcal{G}_B\|^2 \right) \\
= & \int d^3 x \, \sqrt{\gamma} \left( \lambda^{-1} Z_E^2 + \lambda^{-1} Z_B^2 + \lambda \mathcal{G}_E^2 + \lambda \mathcal{G}_B^2 \right).
\end{aligned}
\tag{134}
$$

Where $\lambda$ is the characteristic length scale of the system ($\sigma$ in this case and $M$ in the black hole simulations). The powers are chosen so that the constraint monitor is dimensionless.

Figure 6 shows the evolution of the constraint norm for different values of the constraint damping parameter. We can see that when the constraints are exponentially damped, the decay rate increases with $\kappa$.

These results give us good confidence that the flat electrodynamics part of the code is well implemented. The terms related to curved spacetime are evaluated in the following section.

## 4.2 Static Reissner-Nordström simulation

In order to test the interaction of gravity and electromagnetism in the simplest case, we set up initial data corresponding to a Reissner-Nordström black hole and check that we have both a convergent and static evolution.

We set up initial data in Kerr-Schild coordinates [50] because these are smoothly horizon penetrating, and we set an excision surface at $r = 1.7M$. The initial data for the electromagnetic fields are

$$
E^i = \frac{1}{\sqrt{1+H}} \frac{Q}{r^2} (\partial_r)^i, \qquad B^i = 0,
\tag{135}
$$

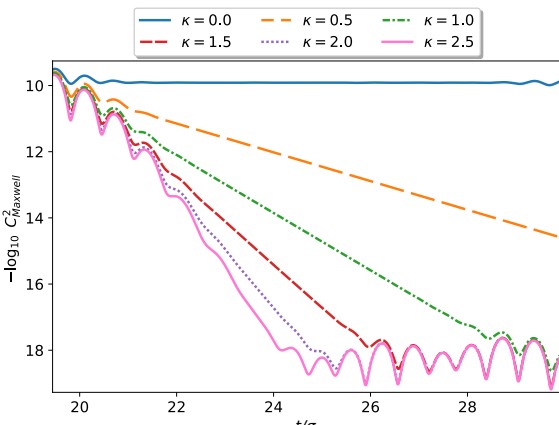

Figure 6: Constraint norm as a function of time $t$ in a simulation of an electromagnetic dipole wave on a flat background for different choices of the constraint damping parameter $\kappa$.

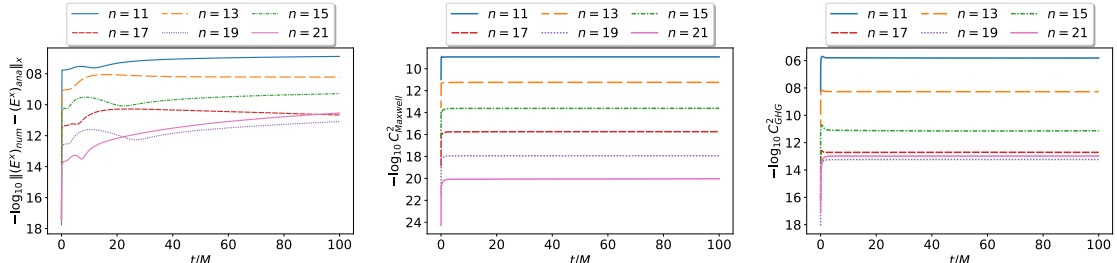

Figure 7: Convergence of various error norms as function of time $t$ with the number of collocation points $n$ for a simulation of a Reissner-Nordström black hole. Some relevant parameters of the simulation are given in table 1. Left: Norm of the difference to the analytic solution taken over the $x$ axis only. Middle: Maxwell constraint norm. Right: Norm of the GHG constraint monitor.

where

$$H = \frac{2M}{r} - \frac{Q^2}{r^2},\tag{136}$$

is the free function that appears in Kerr-Schild coordinates. The initial data for the gravitational field are given by

$$ds^2 = -(1-H)d\tau^2 + (1+H)dr^2 + H(drd\tau + d\tau dr) + r^2 d\Omega^2,\tag{137}$$

$$K_{ij} = \frac{1}{2}\frac{H+2}{\sqrt{H+1}}\partial_r H(dr^2)_{ij}.\tag{138}$$

As stated above, we employ a first order reduction of the GHG formulation of GR. In this experiment, we chose gauge source functions that correspond to the lapse and shift of a Reissner-Nordström spacetime in Kerr-Schild coordinates, which have zero time derivative in the initial data.

For the convergence test we again keep the number of patches fixed and vary the number of collocation points. Figure 7 shows the convergence to the analytical solution $E_{\text{ana}}^{\phi}$, the convergence of the Maxwell system constraint norm $C_{\text{Maxwell}}^2$, and the convergence of the GHG constraint monitor with respect to the number of collocation points. The GHG constraint monitor was introduced in [14] and quantifies both the Hamiltonian and Momentum constraints

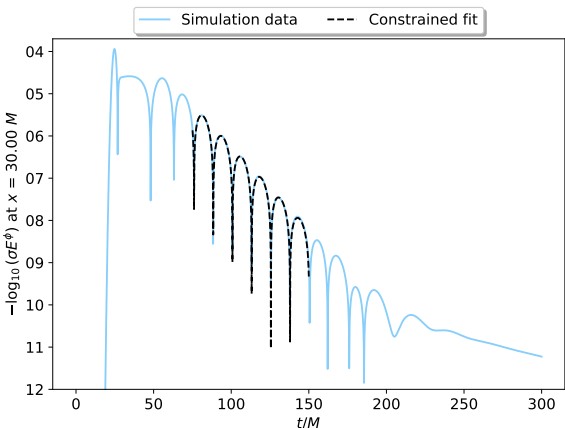

Figure 8: Comparison of the simulation data with the corresponding first quasinormal mode in a Kerr black hole. We can see good agreement.

as well as the violation of the order reduction and harmonic constraints. The figure shows that the error is approximately static, and that the solution converges exponentially with resolution.

## 4.3 Relaxation of an electromagnetically excited black hole

In this test, we set up initial data that is conformally Kerr, with an electromagnetic excitation, and measure the frequency of the fundamental mode of relaxation. It is a fully non-linear simulation in which the geometry is evolved at the same time as the electromagnetic field. For the initial data, we use the CTS solver with freely specifiable variables set to those of a Kerr black hole. For the electromagnetic variables, we choose

$$\tilde{E}^i \partial_i = \mathcal{A}e^{-(r-r_0)^2/\sigma^2}(-y\,\partial_x + x\,\partial_y)\,, \qquad \tilde{B}^i = 0\,. \tag{139}$$

These are analytic solutions to the electromagnetic constraints in the conformal Kerr geometry, since the electric field only points in the azimuthal direction and the spacetime is axisymmetric and hence

$$\bar{D}_i \tilde{E}^i = \partial_\varphi \tilde{E}^\varphi + \frac{1}{2}\tilde{E}^\varphi \partial_\varphi \log \bar{\gamma} = 0\,. \tag{140}$$

We expect the end-state of evolution to be a Kerr black hole. In particular, one might wonder if the electromagnetic excitation would lead to Kerr-Newman instead, however there is no charge in our initial data, and that must remain so during evolution. After the initial data are set up, the evolution is performed until $t = 250M$, with the boundaries at $r = 500M$, since we find that this simulation is very sensitive to boundary effects. The results are presented in Fig. 8, where we compare them with the analytical expectation of the first quasi-normal mode around a Kerr black hole. The frequency to compare is taken from [51–53]. We find good agreement of the simulation and the theoretical expectation.

## 4.4 Electromagnetic multipoles in a dynamical spacetime

As a last test, we evolve a strong gravity case with electromagnetic multipole initial data. In this evolution, the electromagnetic field interacts with itself through gravity and ends up dispersing. This is a good representation of the type of simulations that are required to study critical collapse, and is a good test for the full numerical setup, with the interaction with AMR being of particular interest.

For the initial data, we set the conformal electromagnetic fields to be the same as in Eqs. (123) and (124), with an amplitude to width relation of $\sigma\mathcal{A} = 0.7$, and solve the gravity

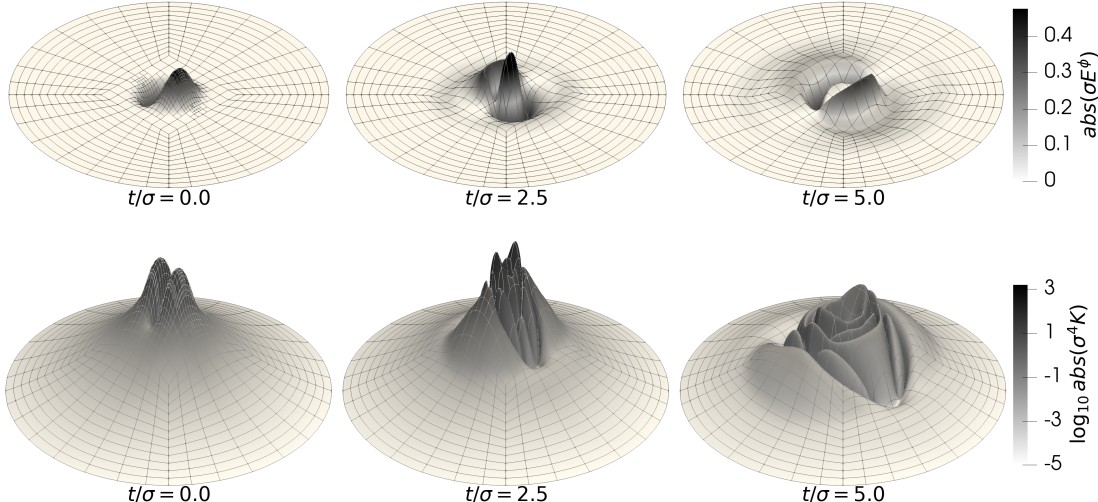

Figure 9: Top panel: Snapshots of $\sigma E^{\phi}$ at different times, in the $xz$ plane for a electromagnetic amplitude of $\sigma \mathcal{A} = 0.7$. Bottom panel: Snapshots of the logarithm of the Kretschmann scalar, as given by Eq. (141), at different times, in the $xz$ plane.

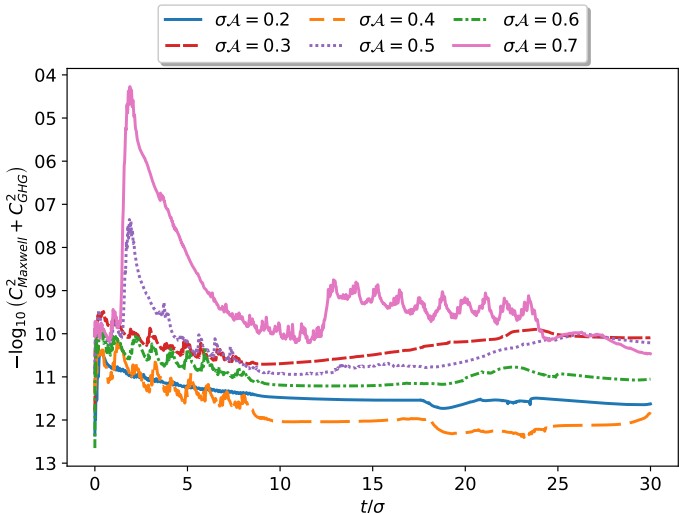

Figure 10: $L^2$-norm of the constraint monitors throughout a fully nonlinear evolution, for different amplitudes of the initial data.

constraints with the choices for the freely specifiable variables corresponding to flat spacetime. This amplitude is not enough to collapse, and is not well tuned to the threshold of collapse, but nevertheless allows us to see comparable strong-field dynamics before dispersing.

The top panel of Fig. 9 shows snapshots of $E^\phi$ in this simulation. Notice the difference to the evolution on a flat background that is shown in Fig. 2. For instance, the electromagnetic field takes longer to disperse, which we can interpret as the effect of self-gravitation that keeps the wave packet together for longer. Based on the knowledge gathered in other matter models [54], we expect this time to increase as we approach the threshold of collapse. The bottom panel of Fig. 9 shows snapshots of the Kretschmann scalar for the same evolution. In particular, we look at its logarithm in units of $\sigma$

$$\log\!\big(\sigma^4 \big|R_{\mu\nu\rho\sigma}R^{\mu\nu\rho\sigma}\big|\big). \tag{141}$$

Even with this poor degree of tuning to the threshold of collapse, we see complicated dynamics in the curvature, which reaches large values before dispersing.

Finally, in Fig. 10 we show the evolution of the constraint monitors. These integrals are several orders of magnitude below the Kretschmann scalar. Consequently, they are small enough and we can trust the simulation results. These measures of error will be more challenging to keep under control with more tuning, but past experience [55–57] indicates that the combination of constraint damping with the BAMPS AMR scheme will go a long way to do so efficiently.

## 5 Conclusions

Motivated by future investigation into various aspects of the electrovacuum dynamics, we have presented a formulation of the Einstein-Maxwell system as an IBVP. We showed that it has several desirable properties for simulations, including symmetric hyperbolicity and boundary stability. Our main goal in choosing both a mathematical formulation and a discretization scheme was to maximize the accuracy to cost ratio.

To the best of our knowledge, we have developed the first PSC implementation of the Einstein-Maxwell system, and this will allow us to obtain accurate results in extreme scenarios. We have presented a suite of tests, demonstrating the correctness of the different parts of the code. An evolution of a dipole electromagnetic wave on a fixed flat background verified the flat terms on the new evolution code. A static Reissner-Nordström simulation verified the interaction with gravity in a simple setting. A measurement of electromagnetic quasinormal modes around a Kerr black hole agreed with the existing literature, thereby validating the physical results of our code. Finally, a strong gravity but subcritical electromagnetic dipole run verified the interaction of the new code with the extensive numerical infrastructure, e.g. adaptive mesh-refinement in a highly dynamical setting. Our conclusion is that the code is working as expected, and is able to evolve the systems of interest.

In terms of the future of the code, an interesting avenue would be to include interaction with a scalar field in order to expand the systems that we can study as well as to provide an interface with recent mathematical work (for instance that of [58]).

The code and the tests here presented provide a good basis to study topics such as critical collapse, but also the dynamics of the Einstein-Maxwell system in more generality, with particular examples being the mode-mixing of electromagnetic waves and gravitational waves in a strong background magnetic field (the Gertsenshtein effect [59]), or the electromagnetic contributions to quasinormal mode ringing.

# Acknowledgments

We are grateful to Florian Atteneder and Fernando Abalos for helpful discussions and to Stephanie Thabuis for revising the text.

**Funding information** J. E. P. and H. R. R. acknowledge financial support provided under the European Union's H2020 ERC Advanced Grant "Black holes: gravitational engines of discovery" grant agreement no. Gravitas-101052587. Views and opinions expressed are, however, those of the authors only and do not necessarily reflect those of the European Union or of the European Research Council. Neither the European Union nor the granting authority can be held responsible for them. D. H. was partially supported by the FCT (Portugal) 2023.12549.PEX grant. The authors were partially supported by FCT (Portugal) projects UIDB/00099/2020 and UIDP/00099/2020. The authors thankfully acknowledge the computer resources, technical expertise and assistance provided by CENTRA/IST. Computations were performed at the cluster "Baltasar-Sete-Sóis" and supported by the H2020 ERC Advanced Grant "Black holes: gravitational engines of discovery" grant agreement no. Gravitas-101052587. The authors gratefully acknowledge the Gauss Centre for Supercomputing e.V. (www.gauss-centre.eu) for funding this project by providing computing time on the GCS Supercomputer SuperMUC-NG at Leibniz Supercomputing Centre (www.lrz.de).

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
