# Peer review of "Pseudospectral implementation of the Einstein-Maxwell system"

_SciPost Physics, doi:SciPost Phys. 19, 112 (2025)_

## Round 1 · Referee Report · Anonymous (Referee 1) · 2025-6-23

Strengths

  1. Very clear and detailed analysis of the Einstein-Maxwell system, including: 1a. well-posedness analysis; 1b. characteristic analysis; 1c. careful treatment of constraint-preserving boundary conditions.
  2. Mostly self-contained.
  3. Very clearly written and well-organised paper.
  4. Very careful analysis of the presented results.

Weaknesses

  1. The work itself and the presented results are not very original.
  2. As far as I could tell, the code that was developed and presented in this paper is not available to the wider community.

Report

This work presents a very comprehensive analysis of the Einstein-Maxwell system and its numerical implementation in a pseudospectral evolution code. While some aspects of this analysis were previously available in the literature, the present manuscript excels in summarizing all relevant details. Notably, it includes a treatment of constraint-preserving boundary conditions and their stability, which is not commonly found. Overall, this work describes the most accurate numerical implementation of the Einstein-Maxwell system that I am aware of.

I have listed some relatively minor points that the authors should clarify in the next section. However, my main criticism of this work is that the code developed, as far as I understand, is not open. In my opinion, this is not a minor detail. Open science has been a significant talking point recently, especially in the context of works funded by the European Union. It also aligns well with SciPost's philosophy of Open Access. And the fact remains that, as it stands, the results from this work are not reproducible since the underlying code is not available to the wider community.

I will leave it to the discretion of the Editors to decide whether this issue constitutes an impediment to publication.

Requested changes

  1. Minor typo in Eq. 4: index $\nu$ on the rhs should be lowered.
  2. The overall infrastructure in which this code is implemented (BAMPS) is mentioned very briefly and only in passing (eg, page 5). While I understand that a detailed analysis is available in Ref [14], the present work would be more self-contained if a brief summary was presented, in particular regarding the (multipatch) coordinates used, excision techniques employed, parallelization, etc.
  3. On page 6, in Eqs. 28-29, it was not clear to me how $E^A$ and $B^A$ are defined.
  4. On page 8, line 211, I was confused with the phrase "incoming constraint radiation". $\phi_0$, as I understood it from Eq. 57, is the actual physical (incoming) electromagnetic radiation, no? What is the "constraint radiation" in this context?
  5. On page 13, line 334, is $\psi$ also freely specifiable data?
  6. General comment on figures 3, 4, 6 and 9: it is not straightforward to see from these plots that the convergence is indeed exponential (as mentioned in the text). While a clear improvement is seen when increasing the number of collocation points, it is not obvious at all from the plots if the convergence is exponential or (high-order) polynomial since there are no expected lines to guide the eye. Also, in figure 3 (left plot): what happens after t>20? Is convergence lost?
  7. On page 19, line 479, excision surface is mentioned but it was never mentioned how this is treated numerically in practice (related with item 2 above).
  8. There is no comment on code runtime, number of CPUs used, efficiency and/or scaling properties of the code, or on how the present PSC implementation compares with other (finite-difference) implementations of this system.

Recommendation

Ask for minor revision

  • validity: high
  • significance: high
  • originality: good
  • clarity: top
  • formatting: perfect
  • grammar: perfect

Author:  Jorge Expósito Patiño  on 2025-09-22  [id 5842]

(in reply to Report 1 on 2025-06-23)

We appreciate the referees comments, and we believe the requested changes clarify both the presentation and results. The changes we have implemented are as follows, by comment index.

We stress the originality of the proof of boundary-stability that we include, since, to the best of our knowledge, for the Einstein-Maxwell system no proof of boundary stability existed for constraint preserving boundary conditions that were employed in the context of numerical relativity. We now make this point clearer in the manuscript.

  • 2 , 3 and 7. To improve clarity we add a new paragraph to explain the infrastructure, including a description of the excision method. We now also give a definition for \(E^A\) and \(B^A\).

  • 4. Regarding the "constraint radiation", the word "constraint" is indeed wrong. \(\phi_0\) is the physical radiation.

  • 5. We now add extra information to clarify the conformal factor \(\psi\) is not freely specifiable, but is to be solved for in the initial data.

  • 6. A new figure is added to show clearly the convergence. In the text we explain that convergence to the analytic solution is lost after \(t/\sigma > 20\) due to the causal contact with the outer boundary. We now added that explanation also to the caption of figure 3, not just in the text.

  • 8. We have added the cost in core-hours of each simulation. Efficiency and scaling of the underlying infrastructure are present in the references. We agree that a code comparison with a different code base would be really illuminating, but we feel it falls outside of the scope of the current paper, as it is something that concerns the entire infrastructure. We will return to this point in future studies.

Lastly, there is the question of the closed source code. The authors of this paper are actively working on the process of preparing the underlying infrastructure (bamps) to be open source in the near future. Given that this is a infrastructure used by many researches, of which the authors don't posses the copyright, it isn't a trivial process, so it most likely won't be finished by the time this paper continues the review process.

Anonymous on 2025-09-27  [id 5867]

(in reply to Jorge Expósito Patiño on 2025-09-22 [id 5842])

I thank the authors for their careful reply. I believe all my comments were addressed.

I have noticed just one minor issue which I had not noticed on my first review. On section 4.1, when the authors present the Flat background test, the initial data is provided in equations 123 and 124. This initial data leads to a dynamical evolution, which is seen in Figure 2. This evolution is then used for convergence testing and in Figure 3 (left) it is shown the convergence of the numerical data to the analytical solution as a function of time. However, this analytical solution is never explicitly written down. For completeness, I believe this would be valuable to have.

Other than the minor issue above, I am happy with the current version of the manuscript and I thank the authors once again for implementing my suggestions.

---

## Round 1 · Referee Report · Nils Deppe (Referee 2) · 2025-7-7

Report

The paper presents a new formulation, implementation, and verification of the Einstein-Maxwell equations. The authors demonstrate stability, convergence, and applicability of the approach to studying critical behavior and quasinormal modes of black holes. The paper is very well written and provides important contributions to the field of numerical relativity. I have a few suggestions for the authors to consider before publishing, but I do not need to review the paper again before it is published.

  • Line 64: It would be good to be explicit about units. Presumably G=c=1, but what about the EM units? With matter there's now a 3rd scale involved and so more units need to be chosen.
  • Line 71: The authors should clearly define the convention for Levi-Civita tensor and symbol that they are using, since there are several different ones used in the literature.
  • Line 79: It would be good to explicitly state and define the Lie derivative, mostly for completeness.
  • Line 81: it would be good to explicitly write out how D_i is defined in terms of partial derivatives, etc. Again, just for completeness and making the paper easier to reproduce!
  • Fig. 2: since the solution here ranges from positive and negative, naively I would've expected a divergent colormap centered at 0. I encourage the authors to do so or to clearly state why they went with the colormap they are using.
  • Line 447: I suggest stating the approximate time the pulse reaches the outer boundary to help guide reader's eyes.
  • Line 445: In case it's helpful, SpEC and SpECTRE also have a very hard time being limited by time stepper accuracy. In BBH simulations this only seems to kick in once spatial errors are driven down to about 1e-13 while the time integrator is using a tolerance of 1e-8. However, this seems to be an artifact of the adaptive stepper not really staying consistently below the CFL, and then rapidly decreasing the time step. spectre even has a scalar wave solution that satisfies the discrete DG equations just to test time stepper convergence: https://spectre-code.org/classScalarWave_1_1Solutions_1_1SemidiscretizedDg.html
  • Fig. 8: I encourage the authors to clearly label the quantities in the figure itself, not just the caption, and to use a different colormap for the Kretschmann scalar to distinguish it further. (And it changes sign, though not clear that's significant)
  • Line 538: possible typo "electovacuum" -> "electrovacuum" (missing "r")?

Recommendation

Publish (easily meets expectations and criteria for this Journal; among top 50%)

  • validity: -
  • significance: -
  • originality: -
  • clarity: top
  • formatting: perfect
  • grammar: perfect

Author:  Jorge Expósito Patiño  on 2025-09-22  [id 5841]

(in reply to Report 2 by Nils Deppe on 2025-07-07)

We are grateful to the referee for their comments, which have pushed us to be more explicit in several places. We have addressed the comments in the revised manuscripts.

---

## Round 2 · List of Changes

Overview of changes requested by referee 1

  1. The typo in Eq. 4 is fixed.

  2. A new paragraph (lines 357-379) is added to explain the infrastructure.

  3. The definition of \(E^A\) and \(B^A\) is added (line 150).

  4. The typo (extra constraint word) is fixed.

  5. Extra information is added to clarify that \(\psi\) is not freely specifiable.

  6. A new figure 5 is added to show clearly the convergence, and the explanation of what happens after \(t/\sigma > 20\) is added also to the caption of figure 3, not just in the text.

  7. The new paragraph that explains the infrastructure (lines 357-379) also includes a description of the excision method.

  8. We have added the cost in core-hours of each simulation.

Overview of changes requested by referee 2

in the same order that the comments were given:

  1. We have added the specifier of Gaussian units, and explicitely added which quantities are set to one in which cases.

  2. The convention is now explicitly stated.

  3. We have added the formula explicitly, both for vectors and scalars. We felt that giving the general formula instead of the two cases separately will be counterproductive if the objective is to be as explicit as possible.

  4. The formula for the spatial covariant derivative \(D_i\) is given explicitly in terms of the four covariant derivative \(\nabla_i\).

  5. The surface plots (now figures 2 and 9) are now colored with respect to the absolute value, which is equivalent to centering at zero with the same color at the maximum and the minimum. We believe this is a good way to color this plot, since the sign of the field is more clear in the elevation of the surface plot.

  6. The approximate time of the pulse reaching the outer boundary is added both in the text, and in the label to figure 3.

  1. For all surface plots (now figures 2 and 9) a label is included next to the color bar. We have kept the same colormap for the Kretschmann scalar and the electric field, for visual consistency, since with the new label the difference between the two is clearer.

  2. The missing "r" that was missing in "electrovacuum" has been fixed.

---

## Editorial Decision

published